# Elucidating the Effects of the Lipids Regulators Fibrates and Statins on the Health Status of Finfish Species: A Review

**DOI:** 10.3390/ani13050792

**Published:** 2023-02-22

**Authors:** Manuel Blonç, Jennifer Lima, Joan Carles Balasch, Lluis Tort, Carlos Gravato, Mariana Teles

**Affiliations:** 1Department of Cell Biology, Physiology and Immunology, Universitat Autònoma de Barcelona, 08193 Barcelona, Spain; 2Department of Physiology, Institute of Bioscience, Universidade de São Paulo, São Paulo 05508-090, Brazil; 3Faculty of Sciences of the University of Lisbon—FCUL, Campo Grande, 1749-016 Lisboa, Portugal; 4Institute of Biotechnology and Biomedicine, Universitat Autònoma de Barcelona, 08193 Barcelona, Spain

**Keywords:** lipid-lowering agents, fibrates, statins, fish, gemfibrozil, atorvastatin

## Abstract

**Simple Summary:**

Pharmaceuticals used to treat abnormal cholesterol levels in the blood are known as lipid regulators (fibrates and statins), and their use is in constant growth. Treatment plants are usually unable to efficiently remove these compounds from wastewater, where their degradation rate is considerably slow, making them an emerging concern for aquatic systems. The present work reviews previously published research concerning the effects of these pharmaceuticals on several finfish species worldwide. Results suggest that both short- and long-term exposure to lipid regulators may have negative effects on fish health, affecting their metabolism and immune system and causing reproductive and developmental disorders. However, the information on these compounds in the available literature is still limited, and additional research is needed to fully understand the threat that their presence in aquatic systems may pose to the production of finfish by the aquaculture industry.

**Abstract:**

The most documented fibrates are gemfibrozil, clofibrate and bezafibrate, while for statins, the majority of the published literature focuses on atorvastatin and simvastatin. The present work reviews previously published research concerning the effects of these hypocholesterolaemic pharmaceuticals on fish, with a particular focus on commercially important species, commonly produced by the European aquaculture industry, specifically in recirculated aquaculture systems (RAS). Overall, results suggest that both acute and chronic exposures to lipid-lowering compounds may have adverse effects on fish, disrupting their capacity to excrete exogenous substances, as well as both lipid metabolism and homeostasis, causing severe ontogenetic and endocrinological abnormalities, leading to hampered reproductive success (e.g., gametogenesis, fecundity), and skeletal or muscular malformations, having serious repercussions on fish health and welfare. Nonetheless, the available literature focusing on the effects of statins or fibrates on commonly farmed fish is still limited, and further research is required to understand the implications of this matter on aquaculture production, global food security and, ultimately, human health.

## 1. Introduction

The daily use of man-made products such as pharmaceuticals, cleaning agents and plastics is in constant and accelerating growth [1], and many of these contaminants have been detected in a variety of environments [2,3]. This uncontrolled anthropogenic activity has triggered several potential risks for aquatic environments [4,5,6] since several contaminants have been quantified in considerable amounts both in surface and ground waters, as well as within organisms [7,8]. Conventional wastewater treatment processes are still unable to efficiently remove all contaminants from influents, and a substantial share of these pollutants, known as contaminants of emerging concern (CECs), remain in the effluent due to their high persistence and low degradation rates [1,9,10,11,12,13]. The input of many compounds classified as personal care products (PPCP), flame-retardants, endocrine-disrupting chemicals, polyfluoroalkyl substances and pharmaceuticals from wastewater treatment plants are not monitored in water, potentially subjecting different organisms to chronic exposures to various chemicals [14,15,16]. Improvements in analytical techniques allow for the detection of an increasing number of these compounds in a variety of bodies of water (e.g., stagnant and running water; surface and ground waters; salt, fresh and brackish water; and wastewater treatment plant influents, effluents and sludge) [17,18,19,20], as well as in food items [16,21]. Pharmaceuticals play a major role in life quality and welfare, but the fast increase in their use has led to significant pressure on natural ecosystems [20,22,23,24]. The over- and misuse of pharmaceuticals, combined with their persistence in the environment, and their proven ability to bioaccumulate on both organisms and sediments [25], make these compounds, and their respective bioactive metabolites, potential chemical stressors to aquatic and cultured fish, and, consequently, a possible threat to humans.

Common aquaculture practices face a number of challenges (e.g., disease outbreaks, feed components shortages, droughts, unexpected flow of contaminants and xenobiotics) and present multiple threats to natural environments, such as eutrophication, habitat destruction and organisms escaping from aquaculture facilities into the wild [26,27,28]. To overcome the effects of those stressors on farmed fish, over the past few decades, recirculated aquaculture systems (RAS) have caught the attention of seafood farmers all over the world and have been increasingly developed at an international scale [27]. RAS are based on the concept of reusing or recirculating, over 90% of the production water, depending on the specific design, therefore minimising water exchange with the surrounding environments compared to traditional aquaculture practices [29,30]. The recirculated water is repeatedly treated with biological, mechanical, and often UV filters in order to maintain appropriate water quality [30,31]. It is now widely accepted that the implementation of RAS provides a wide range of benefits and that this technology may be increasingly adopted to raise all fish species. Firstly, the considerable reduction in water use allows for RAS to be installed in a variety of environments, including areas where water is scarce [31]. Second, being a virtually closed system, RAS enable the farmers to hold full control of water parameters that could naturally fluctuate in a body of water, such as oxygen levels, the concentration of nitrogenous compounds, temperature, and, eventually, the presence of pathogens [30,32,33].

In addition, RAS are usually installed indoors, therefore being sheltered from external factors such as climatic events and the potential predation of cultured organisms from wild animals [27]. Last, but not least, RAS significantly reduce the impact of aquaculture farms on neighbouring natural ecosystems. Indeed, by isolating the facility from the surrounding environments, the probabilities of eutrophication, escapees with potentially deleterious effects on wild populations, pathogen transfer or contamination outflow is minimised [27,30]. The low rates of water exchange, which directly translate to a considerable reduction in water usage, prevent surrounding rivers or groundwater to be heavily polluted or depleted as a consequence of this activity [28,33], and minimise the risk of pathogens entering the systems, translating to a decreased use of antibiotics [31,34].

These factors, in conjunction with the versatility of RAS regarding the species and life stages suitable for culture in such systems [27,31], as well as the different possible stocking and production densities [30], make RAS an appropriate approach to deal with the growing food demand while minimising the environmental impact of the food production industry [27,30,35]. Nonetheless, the array of technicalities incurred by RAS entails farmers developing further skills and expertise and must therefore undertake specific training in order to properly manage the entire system [30]. Furthermore, the different components of RAS, including power supply and sensory systems, are particularly expensive, and consequently, it has been observed that these constraints may restrict the development of these aquaculture practices to areas with greater financial power, such as European countries, North America or some Asian countries [27,36,37,38]. Therefore, further technological advances are required to make RAS truly cost-effective, and to enable its expansion and implementation all over the world [27]. In Europe, RAS farms are principally used to culture aquatic invertebrates (e.g., molluscs and crustaceans), and although the vast majority of farmed fish are salmonids (i.e., *Salmo trutta*, *Oncorhynchus mykiss*), other fish such as *Anguilla anguilla* (European eel) are also widely produced [37].

As pharmaceuticals are now recognised as ubiquitous in aquatic systems [39], and with treatment plants proving unable to remove these from influents [40], it can be assumed that these contaminants are also present in RAS and will affect cultured fish, and consequently aquaculture production. Here we will focus on the effects of lipid regulators, a group of pharmaceuticals engineered to treat dyslipidaemias in humans, acting as lipid-lowering agents, and as primary prevention methods for cardiovascular diseases [39,41].

Lipid regulators are usually manufactured from a variety of compounds such as atorvastatin, bezafibrate, clofibrate, ezetimibe, fenofibrate, fluvastatin, gemfibrozil, lovastatin, mevastatin, pravastatin, rosuvastatin or simvastatin. The first drug of its kind introduced to the market was clofibrate, followed by bezafibrate, fenofibrate and gemfibrozil [42]. Due to the rising need for anti-cholesterol treatments, with earlier-stage treatments and higher dosages [43], lipid regulators are widely prescribed all around the world, and consequently, comprise some of the most reported pharmaceuticals in drinking and wastewater [42,44,45]. The prescription and consumption of lipid regulators multiplied between 2000 and 2017 in countries members of the OECD, with the United Kingdom, Denmark and Belgium displaying the highest consumption rates [42]. Lipid regulators can be sub-classified into statins and fibrates [39], and, even if both are ubiquitous in water bodies, fibrates are often found at higher concentrations [42].

The present work aims to review the currently available literature on the presence of lipid-regulating agents (i.e., fibrates and statins) in the aquatic environment, and their effect on fish, including those commonly farmed in European RAS.

### 1.1. Fibrates

Fibrates decrease levels of fatty acids, triglycerides and low-density lipoproteins, mainly by stimulating the peroxisomal, and partly the mitochondrial, β-oxidation pathways, lowering blood cholesterol levels [39]. On the other hand, fibrates act as agonists of the peroxisome proliferator-activated receptors alpha (PPAR-α), expressed in the liver, heart and skeletal muscle [46]. In the literature, the most commonly investigated fibrates are gemfibrozil, bezafibrate and clofibric acid, with the latter being the most commonly detected in drinking water and food [42]. The most common lipid regulators in water bodies, namely, atorvastatin, bezafibrate, clofibrate, gemfibrozil and simvastatin, have been described to have similar effects in aquatic organisms, particularly vertebrates, as they do in humans, including the induction of changes in lipid levels [42]. Hence, marine and freshwater organisms are vulnerable to the potentially deleterious effects that the presence of such contaminants might incur, making this an issue of major ecological concern. It is known that exposure to lipid regulators generally present in water (e.g., wastewater treatment effluents and drinking water) can have a range of consequences on fish, affecting physiology, metabolism, reproduction, behaviour and immunity [44,47,48,49]. For instance, previous studies have shown that exposure to gemfibrozil may lead to decreased levels of cholesterol in plasma [48], and trigger the organism’s antioxidant response, as well as tamper with a fish’s capacity to efficiently swim in counter-current [47]. Moreover, gemfibrozil, like many other compounds, can bioconcentrate in different tissues in fish [50], potentially having deleterious effects on human health through the ingestion of contaminated food. Gemfibrozil, is resistant to photodegradation, has a long half-life in water [51] and is not completely removed by wastewater treatment plants (WWTP) [40]. In Spain, for instance, removal rates of this pollutant from WWTP ranged between 10–75% [52]. Its persistence in water allows us to find concentrations up to 4760 ng/L in effluents [53], and up to 758 ng/L in coastal waters [54]. Moreover, values ranging between 6.69 and 10.34 ng/L were recorded in Portuguese surface waters [55] and up to 0.8 and 1.7 ng/L in Italian tap and surface waters, respectively [56]. In comparison, in Spain, values of up to 70.27 and 77.8 ng/L were detected in different sites of the Llobregat river [57,58].

Bezafibrate was detected in wastewater treatment facilities effluents in Spain at concentrations ranging between 2 and 132 ng/L, and in estuarine environments at concentrations ranging between 2 and 67 ng/L [59]. In comparison, Portuguese surface waters were contaminated with values of bezafibrate ranging from 11.86 to 15.52 ng/L [55]. Moreover, bezafibrate, gemfibrozil and clofibric acid, an active metabolite of clofibrate, have been detected in groundwater in Barcelona, being found in concentrations of maximum 25.8, 751 and 7.57 ng/L, respectively [18]. Nonetheless, the most predominantly detected fibrate in the aquatic environment is clofibric acid [60,61].

### 1.2. Statins

Statins lower cholesterol levels through the suppression of its biosynthesis as a result of competitive inhibition of 3-hydroxy-3-methylglutaryl coenzyme A reductase (HMG-CoA reductase), which is responsible for cholesterol biosynthesis in the liver [39]. Statins currently available on the market include atorvastatin, cerivastatin, fluvastatin, lovastatin, pitavastatin, pravastatin, rosuvastatin and simvastatin [62,63]. However, data on the toxicity of statins on different organisms are very limited and are generally restricted to atorvastatin and simvastatin, the two most common statins [63,64]. Few studies have focused on the environmental concentrations of statins despite the evident increase in their consumption. Statins appear to have more stable removal efficiencies compared to fibrates, but their presence has already been reported in both untreated and treated sewage water samples at values ranging from 4 to 117 ng/L and 1 to 59 ng/L, respectively [65]. Atorvastatin, for example, was found in low amounts, although its metabolites, namely p-hydroxy atorvastatin, and o-hydroxy atorvastatin, were found in greater amounts in wastewater. A similar trend occurs with simvastatin, where this statin was not directly detected in surface water and sediments unlike its specific metabolite (i.e., simvastatin hydroxy carboxylic acid) and was found up to 108 ng/L in Norwegian cities [66].

The inefficiency of wastewater treatment plants when it comes to the removal of these pharmaceuticals is translated by an inevitable exposure of both wild and cultured aquatic animals to these contaminants of emerging concern, even in closed systems, such as recirculated aquaculture systems (RAS), where they might accumulate in water. In both wild and laboratory-reared fishes, the consequences of exposure to fibrates and statins have been studied to varying extents, depending on compound and species, indicating the potentially deleterious effects of these contaminants on the overall health and welfare of fish (Figure 1). Here, we aim to compile existing knowledge on the effects of lipid-regulating agents, their environmental distribution, as well as their effects on fish, since these pharmaceuticals are likely to influence their lipid metabolism and nutritional quality, with a special interest in organisms that are highly relevant to aquaculture, and, therefore, directly linked to global food security and human health. We will also focus specifically on the most documented drugs, namely gemfibrozil, clofibric acid (a metabolite of clofibrate) and bezafibrate, representing the fibrates, and atorvastatin and simvastatin representing the statins.

## 2. Methodology

Articles published from 2011 onwards were selected in July–October 2022, using “lipid regulators”, “lipid-lowering agent”, gemfibrozil”, “clofibrate”, “clofibric acid”, “bezafibrate”, “fibrates”, atorvastatin”, “simvastatin” and “statins” as keywords. Moreover, the combinations of the aforementioned keywords, in conjunction with “effects in fish”, “aquaculture” and both the common and scientific names of the main species of interest for the European aquaculture industry (i.e., *Salmo salar*, *Oncorhynchus mykiss*, *Anguilla anguilla* and *Clarias gariepinus*) were also used. Due to the scarcity of recent studies focusing on the environmental presence and distribution of these pharmaceuticals in European countries, as well as their effects on the previously specified species of interest to the present work, articles related to these matters were selected, even if the publication was dated prior to 2011.

## 3. Effects of Lipids Regulators on Fish

An increasing number of studies focus on the potentially deleterious effects that the presence of lipid-regulating agents may have on non-target organisms. These suggest that lipid regulators can act similarly in mammals (e.g., humans) and other vertebrates, such as fish, when exposed to these compounds through contaminated waters. This factor, combined with the high use of these drugs, their incorrect discard and their low removal rates, imply a major environmental problem.

### 3.1. Fibrates

#### 3.1.1. Gemfibrozil

A considerable quantity of the published literature has focused on the effects of gemfibrozil in aquatic organisms as it is now an issue of high concern (Table 1). In humans, gemfibrozil is capable of increasing serum aminotransferase, and, in isolated cases, of causing liver damage. The reduction in lipids is associated with the peroxisome proliferator-activated receptor (PPARα), involved in the gene expression of enzymes for the oxidation of fatty acids [45]. In fish, as in other animals, an increase in levels of lipoprotein lipase can occur, subsequently increasing triacyl glyceride-rich lipoproteins [51]. *Danio rerio* (zebrafish) exposed to gemfibrozil through diet for 30 days displayed decreased plasma levels of triglycerides and overall cholesterol levels [44]. These results corroborate the idea that gemfibrozil can operate in similar ways in fish as in humans, notably generating changes in lipid homeostasis and altering the availability of energy resources. Nonetheless, a study with fathead minnow (*Pimephales promelas*) showed an inconsistent effect of chronic exposure to gemfibrozil on lipid metabolism, steroidogenesis and reproduction, and no effect when employing environmentally relevant concentrations [48].

In gilthead seabream (*Sparus aurata*), gemfibrozil was found to affect swimming behaviour [47,90], with the ability to swim in counter-current for extended periods of time decreased by up to 65%. These behavioural effects were further associated with a significant increase in both catalase (CAT) and glutathione reductase (GR) activities in the gills of fish exposed to concentrations of the pharmaceuticals [47]. Furthermore, exposure to gemfibrozil appeared to have genotoxic effects on this species, causing nuclear anomalies [74]. In addition, exposure to gemfibrozil did not change gill total antioxidant capacity (TAC) but increased total oxidative status (TOS) and altered mitochondrial RNA of certain genes such as glutathione peroxidase 1 (GPx) and interleukin 1β in both gills and head kidney of the same species [49]. The transcription of key genes involved in lipid homeostasis was also affected by exposure to gemfibrozil, being characterized as a stress-inducing agent. Furthermore, exposure of *S. aurata* to 15,000 ng/L of gemfibrozil induced the activation of the inflammatory response in the liver [72]. Altogether, these studies suggest that, in *S. aurata*, this compound alters biochemical and gene functions involved in the immune and oxidative stress responses, having serious implications for the health and welfare of this species.

The bioaccumulation of gemfibrozil in both muscle and adipose tissue of juvenile rainbow trout (*Oncorhynchus mykiss*) following an 8-day waterborne exposure to 3000 ng/L of gemfibrozil has been previously demonstrated using space-resolved solid-phase microextraction (SR-SPME) [50]. Prindiville et al. [45] investigated the effects of exposure to gemfibrozil on the lipoprotein metabolism of *O. mykiss.* The authors injected female individuals with 100 mg/kg of gemfibrozil every 3 days over a 15-day period and reported a significant decrease in the concentration of plasma lipids, as well as in lipoprotein concentrations. Such results might translate into a hindered ability to reproduce, to efficiently undergo homeoviscous adaption to changes in water temperature, as well as tampered locomotor capacities, impeding long-distance, constant swimming behaviours [45]. Nonetheless, this is mostly speculative work and required additional research to be carried out to confirm these hypotheses. Furthermore, it was reported that exposure to pharmaceutical doses of gemfibrozil significantly reduced the concentration of n-3 fatty acids, which is likely to have direct repercussions on the nutritional quality of the fish [45,91]. Gagné et al. [92] further corroborated the presence of gemfibrozil in municipal wastewater treatment plants effluent in Canada, once again emphasizing the importance of studying the effects of emergent contaminants on the health of both wild and cultured fishes. However, the authors classified this specific drug as of little-to-no concern to fish, as they observed minimal toxicity of this pharmaceutical in rainbow trout hepatocytes. This is in accordance with previously published findings, also indicating that gemfibrozil had no significant effect on the hepatocytes of *O. mykiss* [93].

On the other hand, Donohue et al. [93] described a significant increase in bioindicators for oxidative stress on hepatocytes, caused by clofibric acid, the metabolite of clofibrate. Similarly, Triebskorn et al. [94] reported a moderate reaction of the liver of rainbow trout to exposure to clofibric acid, including dilation of blood vessels, and no significant reaction in the kidney of the exposed fish. However, results indicated a more severe response from the gills of *O. mykiss* to this pollutant, as the authors described epithelial lifting, increased cell production (hyperplasia) and growth (hypertrophy) of mucus cells. Nonetheless, these reactions are non-specific, as they tend to occur when the gills enter into contact with waterborne contaminants and are often regarded as a measure to preserve ion balance [95]. In addition, through an experiment focusing on the exposure of primary hepatocytes of rainbow trout to clofibric acid alone and combined with female sex hormone, it was demonstrated that this pharmaceutical is not a significant PPAR activator in hepatocytes in this species. In addition, neither lipid metabolism nor estrogenic activity resulted altered from this exposure [96].

The available information on the impact that exposure to lipid-lowering compounds has on the European eel (*Anguilla anguilla*) is extremely limited. According to Lyssimachou et al. [69], gemfibrozil is not a significant PPARα activator in this species. Nonetheless, injections of gemfibrozil significantly decreased the activity of certain pathways directly linked to cytochrome P450, which are responsible for the metabolism of xenobiotics, as well as the oxidation of fatty acids. This could indicate a tampered ability of the fish to efficiently extract energy from their food source, and to develop and function efficiently [97], having potentially serious implications for aquaculture production.

In zebrafish, chronic exposure to low concentrations of gemfibrozil can have transgenerational effects [77]. In adults, reproductive ability is reportedly altered, accompanied by histological changes in ovaries and kidneys. Furthermore, adult zebrafish chronically exposed to 500 and 10,000 ng/L of this pharmaceutical displayed a decrease in reproductive output, with atretic oocytes and altered kidney histology [67,68]. On the other hand, unexposed offspring of parents exposed to this contaminant display sex-dependent alterations in courtship activity and swimming behaviour, and abnormal sperm morphology, significantly affecting their fecundity and breeding success [70], although this seems limited to the first generation of offspring only [77].

Similarly, in a long-term exposure (i.e., 155 days) experiment with Japanese Medaka (*Oryzias latipes*), a significant decrease in fecundity was observed, tied with significant decreases in both testosterone levels in female fish and hatchability of F1 fish. Moreover, the expression of genes related to oestrogen receptors and vitellogenin was increased in both gonads and livers [78]. This suggests that, in some species, gemfibrozil can act as an endocrine disruptor, affecting reproduction and ontogeny. In male individuals of the same species, short-term exposure to gemfibrozil lowered levels of 17β-estradiol, 11-ketotestosterone and plasma cholesterol [78]. The authors argued that the endocrine disruption observed as a consequence of the challenge might be directly linked to the hypocholesterolaemic nature of the medicine [78], altering the reproductive success of individuals, and having varying effects depending on the exposure duration.

The inconsistency in the published results is likely linked to the dependence of the gemfibrozil’s mechanisms of toxicity to exposure duration, administration method, species and life stage, as well as organ analysed. However, it appears as if exposure to gemfibrozil may have strong effects on reproduction (e.g., fecundity, gametogenesis), excretion of exogenous substances (i.e., detoxification process), lipid metabolism and the overall homeostasis. In addition, the presence of this pollutant in aquatic systems is a potential factor affecting larval development, leading to ontogenetic and behavioural anomalies in both adults and larval stages, and, to some extent, in the subsequent generations.

#### 3.1.2. Clofibrate and Clofibric Acid

Clofibric acid, the main metabolite of clofibrate, is a PPAR-α agonist that increases beta-oxidation and decreases triglyceride secretion in humans, but it is known to affect aquatic organisms [60,86]. Even though some of the previous literature investigated the consequences of exposure to clofibrate itself, a major part of the studies hereby considered focused on the effects of clofibric acid (Table 1).

In grass carp (*Ctenopharyngodon idellal*), it was observed that clofibrate decreased triacylglycerol, plasma cholesterol and overall lipid concentrations. Similarly, it increased hepatic enzymes involved in the synthesis of fatty acids, as well as lipoprotein lipase expression. Furthermore, a decrease in lipogenic enzymes, previously increased through feeding with a high carbohydrate diet, was observed [81]. This suggests that clofibrate, like gemfibrozil, can act as a hypolipidemic agent and alter the lipid metabolism of some fishes.

In zebrafish, exposure to clofibrate through ingestion leads to a modulation of genes such as fatty acid-binding proteins (fabp) and acyl-CoA oxidase 1 (acox1), a marker of PPAR-α activation in many tissues. Nonetheless, this process proved to be tissue dependent. Clofibrate is potentially unable to cross the blood–brain barrier of zebrafish, hence the lack of significant alterations in the transcription of acox1 in the brain [80]. In zebrafish larvae, the exposed group displayed shorter body lengths compared to the control group, in addition to changes in morphological characteristics and lethargic behaviour. When fed with this compound, an upregulation of peroxisomes in liver and heart mitochondria in was observed [81], suggesting that waterborne exposure to clofibrate affects both the metabolism of adults and larvae, and may cause strong alterations in the development of embryos, leading to severe deformities and behavioural changes. On the other hand, it has been reported that clofibric acid can cause behavioural and metabolic alterations in adults and larvae of *D. rerio* both after acute and chronic exposures but does not act as an endocrine disruptor. This drug can influence the enzymatic activities involved in counteracting oxidative stress, like superoxidase dismutase (SOD), CAT and GPx, although the level of alterations depends on the pollutant’s concentration. In addition, it has been found that this compound may cause alterations in biotransformation enzymes, as well as in lipid peroxidation levels [88]. Zebrafish embryos have a high potential for metabolizing xenobiotics, involving a variety of enzymatic activities, and generating derivate products such as clofibric acid [87]. It has been previously stated that a lifelong exposure of zebrafish to clofibric acid significantly affected the reproduction, spermatogenesis, growth and gene expression of fabp, in a transgenerational manner [86]. It was determined that the effects are dependent on the individual’s sex, and on the organ considered, and can express inter-generational variations, influencing fecundity, mortality and the occurrence of skeletal deformities.

With the purpose of investigating key enzymatic activities with involvement in reproduction, an in vitro study in common carp (*Cyprinus carpio)* showed that clofibrate had an inhibitory effect upon CYP17, related to a synthesis of active androgens on gonads [79]. In an experiment challenging *C. carpio* the results showed alterations in haematological parameters (decreased red blood cells count and increased white blood cell counts), biochemical endpoints (increased levels of glucose and proteins), ion regulation (decreased plasmatic sodium, increased Na^+^/K^+^ ATPase in gills) and on enzymatic responses (i.e., Lactate dehydrogenase, LDH; glutamic-oxaloacetic transaminase, GOT; and glutamate pyruvate transaminase, GPT) [82].

In a different study from the same research group, *Cirrhinus mrigala* was exposed to the same concentrations resulting in alterations in enzymatic parameters, such as GOT and GPT after short-term exposure, and changes in the ion homeostasis of gills (Na^+^/K^+^ ATPase activity) after both short- and long-term exposures to this clofibric acid [83]. On the same species, clofibric acid influenced biochemical parameters, causing an increase in plasmatic glucose and Na^+^, and K^+^, as well as a decrease in plasma protein and Cl^−^ following short-term exposures. On the other hand, long-term exposure to this contaminant lead to increased levels of plasmatic Na^+^ and Cl^−^ and oscillations in the levels of plasma glucose, protein and K^+^ [84]. Through a similar experiment, the authors also reported changes in thyroid hormones in the plasma of *C. mrigala* [85].

The published literature focusing on the effects of these lipid-regulating agents on *S. salar* is extremely scarce. Rørvik et al. [98], briefly described a significant decrease in lipid content in Atlantic salmon following ingestion of a feed complemented with clofibrate when compared to a control group.

Overall, these studies suggest, to a certain extent, inter-species variation regarding the effects of clofibric acid. However, the consequences of either chronic or acute exposure to this pharmaceutical often include strong alterations in lipid metabolism and homeostasis, ontogenetic disturbances, and effects on behaviour, haematological parameters and the detoxification process. Furthermore, and similarly to gemfibrozil, clofibric acid has been described as being, to a certain extent, an endocrine disruptor, and as having effects on fish displaying inter-generational and sex-dependent differences. Finally, it appears evident that rising environmental concentrations of this pollutant may have serious implications for the health and welfare of wild fish populations.

#### 3.1.3. Bezafibrate

Despite being one of the first lipid regulators to be introduced in the pharmaceutical market, there are not many studies investigating the effects of bezafibrate on aquatic organisms when compared to other fibrates, such as gemfibrozil and clofibrate, or its metabolite, clofibric acid. To the best of the authors’ knowledge, the only article available that investigates the effects of this compound reported that exposure to bezafibrate through diet can act like an endocrine disruptor in male zebrafish. This drug can lower plasma cholesterol, decrease 11-ketotestosterone (11-KT), and induce changes at the histological and molecular level in the testis. In addition, the expression of a variety of genes was affected, being either up- or down-regulated depending on exposure time (Table 1). Therefore, bezafibrate affects the gonadal steroidogenesis and spermatogenesis of this organism, consequently having strong implications on the reproduction of the species and affecting it at the population level [89].

#### 3.1.4. Overall Effects of Fibrates in Fishes

It is clear from the literature reviewed that the effects of fibrates in fishes depend strongly upon the still unresolved influence of species-specific physiologies and lifecycles, developmental stage, dosage differences across studies and the amount of time exposed to environmentally relevant concentrations of fibrates. To date, gemfibrozil seems to be the focus of most studies, but overall, the main deleterious effects of fibrates on fish physiology seem to be restricted primarily to alterations in cell and tissue lipid metabolism. This, in turn, may affect the timing and functionality of the reproductive axis and vitellogenesis and may result in abnormalities during the larval development, affecting normal swimming, and perhaps foraging, performance. From the data reviewed, it is not clear if these effects impair the immune responses in adults, but the few reports that indicate alterations in the expression of genes related to inflammation processes suggest that fibrates do not damage long-term immune reactivity. The antioxidant response associated with exposure to fibrates seems, in this sense, mostly related to aerobic metabolic outbursts as a byproduct of lipid oxidative alterations rather than to energy-consuming inflammatory responses. However, this may not be the case in cultured fishes, even in more advanced RAS systems, fed with lipid-enriched formulations, prone to elicit oxidative stress and inflammatory responses [99].

It seems safe to assume that a short-life antioxidant response may dysregulate the metabolic budget during development but not the lifelong (adaptive) performance of adults. However, it is particularly worrying that some of the effects on lipid metabolism can be transgenerational as mentioned in zebrafish studies [77]. Although chronic exposures are hard to reproduce in non-model, free-living species, in fishes, the long-term impact of stressors (including pharmaceuticals, endocrine disruptors and persistent changes in physical variables such as temperature), at the population level seems to rely, partly, on the transgenerational persistence of stress-related disruptive effects in the thyroidal-, estrogenic/androgenic- and growth-related axis [100,101,102,103] all of them highly sensitive to alterations in the metabolism of lipids. Moreover, sex plasticity in fishes depends on the concerted influences of lipid-, temperature-, seasonality- and stress-responsive gene pathways regulating gonadal maturation and fate [104,105,106]. Pharmaceutical-related alterations in the availability and amount of lipid substrates may result in an impairment of fertility output and biases sex ratios.

Of special concern are the changes of Na^+^/K^+^ ATPase transporters and activity resulting from exposure to fibrates in *Cyprinus carpio* and *Cirrhinus mrigala* [82,83]. In salmonids, these changes have still not been described, but if present, they may interfere with the osmoregulatory adjustments (smoltification) required for the migratory phase of diadromous fishes, in which branchial mitochondrial-rich ionocytes and the management of hepatic lipid stores determine the onset of smoltification [107], a complex process still understudied. In Atlantic salmon, prior to sea migration, lipid metabolism becomes less plastic to changes in lipid diets as demonstrated by the variations in the expression of genes linked to lipid uptake in the gut and hepatic lipogenesis and lipid transport [108]. These shifts in the metabolism of lipids are life-stage dependent, and constrict the growth rate that, in turn, acts as a trigger to undergo, or not, smoltification. In this sense, it would be worth studying the effects that exposure to fibrates during phases of major osmoregulatory and morphological changes may have on the obligate or facultative development of smoltification in migrant salmonids, a process that requires behavioural changes and modifications of swimming performance that may be influenced by the presence of gemfibrozil in the brain circuitry (see Figure 1).

### 3.2. Statins

Statins inhibit an enzyme called 3-hydroxy-3-methylglutaryl CoA reductase (HMGCR) that converts 3-hydroxy-3-methylglutaryl-CoA (HMG CoA) into the cholesterol precursor mevalonate. Thus, statins decrease cholesterol synthesis as they compete with HMG CoA for the active site of the enzyme, altering its conformation and inhibiting its function [109].

Although waterborne statins are expected to be bioavailable to fish due to their high input rate and persistence, there are indications that they will not bioaccumulate in organisms [25,50]. In addition to uptake via passive diffusion through membranes, statins may be taken up and excreted by membrane transporters. The activity and expression of membrane transporters in the gills of aquatic organisms may therefore influence their bioavailability by affecting their influx and/or efflux rates [110]. Therefore, the effects of this contaminant on an aquatic organism may vary greatly between species, or life stages within the same species (Table 2), but might be less severe than those observed for fibrates.

#### 3.2.1. Atorvastatin

The prescription rate of statins continues to increase, mainly with the high sale of atorvastatin, often under the brand name “Lipitor” [44,124], representing a pioneering treatment for hypercholesterolemia [125] but increasing the concentration of this pollutant in aquatic systems.

Exposure to atorvastatin in rainbow trout was reported to alter genes involved in membrane transport, oxidative stress response, apoptosis and metabolism in gills at low concentrations. However, no effect was observed on genes involved in cholesterol biosynthesis or peroxisomal proliferation [111]. In contrast, an experiment focusing on primary rainbow trout hepatocytes showed that this environmental pollutant may alter lipid synthesis and reduce cholesterol biosynthesis [113].

In zebrafish, exposure to this emergent contaminant was directly linked to alterations in cholesterol metabolism, lipid regulation, and steroid production. Specifically, an observed reduction in cholesterol was associated with decreased levels of cortisol and sex steroids. Moreover, a reduction in triglyceride was associated with changes in mRNA levels involved with lipid regulation [44]. Additionally, exposure to this compound resulted in signs of skeletal muscle breakdown, indicating rhabdomyolysis, and strongly tampering with the welfare of male fish. Interestingly, this effect on skeletal muscle seemed to be gender-specific, as different response between male and female was observed [44].

Besides the effects observed on adult individuals, a strong, dose-dependent, myotoxic effect of this drug was reported in zebrafish larvae [112]. In addition, exposure to atorvastatin significantly reduced the response to tactile stimuli, and caused a decrease in enzyme activity, like citrate synthase (CS) and cytochrome oxidase (COX). This suggests involvement in metabolism in larvae, thus, affecting the early development of the fish, and, ultimately, success in adults. [112].

The effects of atorvastatin seem somewhat similar to those described for fibrates, with important implications for an individual’s development, and lipid, as well as overall metabolism, also acting as an endocrine disruptor.

#### 3.2.2. Simvastatin

As with the other lipid-regulating agents mentioned before, exposure to simvastatin has been reported to have a variety of consequences in fish. For instance, exposure of fewer than 7 days was able to cause significant alterations in the antioxidant system of mosquitofish (*Gambusia affinis*) and related enzymatic activity, such as nuclear factor erythroid 2–related factor 2 (Nrf2) and mitogen-activated protein kinases (MAPK) [122]. Furthermore, the presence of this contaminant in aquatic environments is reportedly responsible for strong histological changes in hepatic cells of the same species [119,122]. Similarly, histological changes were reported to occur in *Mugilogobius abei* following short-term exposure to this statin. Furthermore, increased expression of genes related to xenobiotic resistance and detoxification (pregnane X receptor, PXR), including cytochromes P450 (i.e., CYP1A, CYP3A), has been shown to be somewhat induced by short-term exposure to this pollutant, concomitantly with alterations in enzymatic activity [120].

In zebrafish, it has been reported that chronic exposure to simvastatin at environmentally relevant concentrations, may have, in certain parameters, both dose-dependent, and sex-dependent consequences [118,121]. Indeed, following a 90-day waterborne exposure to this contaminant, the authors observed significant differences in the expression of genes responsible for energy metabolism (e.g., fatty acids synthesis and metabolism, glucose metabolism) in the brain [121]. Further analyses performed during the same experiment indicated significant alterations in the mevalonate pathway (e.g., directly involved in the synthesis of cholesterol), as well as a sex-dependent variation in body weight and body length at the highest exposure concentrations. However, a significant increase in reproductive success (fertilisation %) was observed at 200 ng/L, although this was followed by an increase in developmental anomalies. In addition, chronic exposure to simvastatin resulted in significant changes in cholesterol and triglyceride levels, although this response was not dose dependent. Moreover, the authors described a significant decrease in the heartbeat rate of embryos following parental exposure to 8 ng/L of this statin [118].

This agrees with what was previously described by Campos et al. [115], who also reported significantly slower heartbeat in zebrafish larvae as a consequence of exposure to simvastatin. Additionally, the authors reported decreased movement and significant structural alterations (e.g., changes in septum shape and size of somites, essential for locomotion in fish). The authors further speculate that exposure to this lipid-regulator may directly affect DNA replication, apoptosis, and muscle function [115]. Zebrafish embryos exposed to high concentrations of simvastatin experienced important mortality, and sever morphological anomalies, compared to low concentrations, which induced barely any mortality, and no morphological alterations [114]. In addition, it has been observed that exposure to this compound caused strong developmental issues, leading to abnormalities in the septum, somites and myofibrils and most likely tampering with the individual’s ability to swim, further corroborating previous findings [114].

On the other hand, Cunha et al. [117] described no significant effects of simvastatin exposure on cholesterol levels of *D. rerio* larvae, although the experimental methods differed with regard to the developmental stages challenged, and the concentrations tested, potentially indicating dose-dependent differences in the effects of simvastatin. Furthermore, the authors reported that exposure to this lipid-lowering agent may cause either up- or down-regulation of nuclear receptors (i.e., PPAR; PXR; constitutive androstane receptor, CAR; retinoid X receptors, RXR), as well as in aryl hydrocarbon receptor (AhR) in zebrafish, depending on the dose of simvastatin, and the exposure duration [117]. In that regard, the authors had previously reported that waterborne exposure to simvastatin could cause severe developmental abnormalities in zebrafish embryos, particularly tail deformities [116].

Moreover, the authors detected significant upregulation of ethoxyresorufin-O-deethylase (EROD), glutathione S-transferases (GST) and CAT as well as different ATP-binding cassette (ABC) transporters and cytochrome P450, indicating the induction of the cell detoxification process, and the related antioxidant procedure. Furthermore, the authors demonstrated the potential for simvastatin to interact with other chemical environmental pollutants, possibly having synergistic effects [116]. In contrast, Rebelo et al. [123] observed a significant decrease in SOD, GST and GPx, as well as thiobarbituric acid reactive substances (TBARS), in zebrafish embryos and juveniles following both acute and chronic exposures to simvastatin, indicating a reduction in antioxidant defences. In addition, significant behavioural changes were detected throughout these experiments, but no alterations in gonad development or sex determination were noted.

#### 3.2.3. Overall Effects of Statins in Fishes

The caveats mentioned above for fibrates are equally relevant for the effects of statins in the physiology and behaviour of fishes, with an amendment: statins are even less studied and in fewer species. Due to the pleiotropic nature of the effects of statins [126], some unrelated to their ability to reduce cholesterol levels, it is hard to say if the intensity and scope of statins’ effects in fish are comparable to those described for fibrates beyond changes in lipid metabolism, abnormal development in larvae and oxidative stress. Either way, even conflicting results seem to indicate no major effects on long-term inflammatory outcomes. However, unexpected systemic alterations related to interferences with metabolic and mitochondrial oxidative pathways should not be ruled out. In mammals, statins have been linked to dose-dependent alterations in angiogenesis [127], and in this sense it would be worth further analysing their putative myotoxic effects during cardiovascular development in fishes [118], widening the focus beyond the classical zebrafish ecotoxicology resources [128] to cover non-model species.

## 4. Concluding Remarks

The manufacturing and use of lipid-regulating agents are constantly growing, and many of these compounds (e.g., statins and fibrates) are ubiquitous in aquatic systems. However, limited information is available on the effects of lipid-regulating agents on the most common species of fish in European RAS, with most research having focused on *O. mykiss*. Nonetheless, the published literature offers a worrying insight into the implications of having aquaculture fishes exposed to such hypocholesterolaemic agents. Indeed, it has been reported that fish might be continuously exposed to these contaminants, and that, although not all these compounds are particularly likely to bioaccumulate, prolonged contact may potentially lead to modified swimming, feeding and social behaviour in these organisms. Furthermore, it has been suggested that lipid-lowering agents may tamper with fish’s ability to reproduce and develop properly, causing deformities if exposed during early life stages, and therefore, greatly affecting the welfare of farmed animals. In addition, preliminary findings indicate that the nutritional quality of cultured fish may be significantly decreased by these contaminants of emerging concern, which may have serious implications for food security, as fish represents a considerable proportion of protein sources worldwide.

The low-level activation of inflammatory responses in fishes exposed to fibrates is intriguing and deserves more attention, considering the mutual influence between lipid and immune tissues [129]. Some regulators of key metabolic pathways, such as the mTOR anabolic signalling pathway, participate not only in the hepatic balance between lipolysis and lipogenesis, but also regulate the onset of inflammatory responses [130]. These pathways, together with the lipid tissue-dependent regulation of local immune responses in fishes [131] enforce the crosstalk between visceral adipose masses and immune cell trafficking and antigen presentation in fishes, similar to what has been observed in mammals.

More chronic studies are needed to ascertain the effects of lipid-regulating agents at the population level in free-living fishes, including those that, such as salmonids, have experienced several rounds of genome duplication, gained a duplicated set of genes related to lipid regulation and management, and, thus, may be more resilient to pharmaceutical-derived metabolic alterations [132]. Seasonal trends in lipid allocation for vitellogenesis and gonadal maturation, coupled with transgenerational alterations due to chronic exposure to environmental stressors have been described in three-spined sticklebacks (*Gasterosteus aculeatus*) in an attempt to dissect the complexity of reproductive allocation of resources during the breeding season [133]. These, and other species with context-dependent lifecycles deserve more studies concerning the synergistic or antagonistic effects of lipid regulators, acting alone or coupled with other stressors, on normal and maladaptive physiology of fishes. In this sense, the activation of the hypothalamic-pituitary-interrenal (HPI) stress axis in fishes responds not only to environmental cues but also to the presence of pharmaceuticals, xenobiotics and other contaminants that may bias the effect of HPI byproducts on the sex plasticity pathways [134], compromising the maintenance of healthy populations.

All these factors suggest that, if the environmental concentrations of statins and fibrates continue to rise, the effects on both wild and cultured organisms will intensify. Therefore, it appears that both the health and welfare of fishes in aquaculture systems will be at risk. Nonetheless, the effects of lipid-regulating agents seem to be somewhat dependent on a variety of factors (e.g., dose, exposure time, sex, and life stage). Thus, although the available literature allows, to a certain degree, for extrapolation of results on farmed species, additional research, with a particular focus on commercially important species (e.g., cultured in RAS), should be carried out to fully understand the implications for aquaculture production and, ultimately, global food security.

## Figures and Tables

**Figure 1 animals-13-00792-f001:**
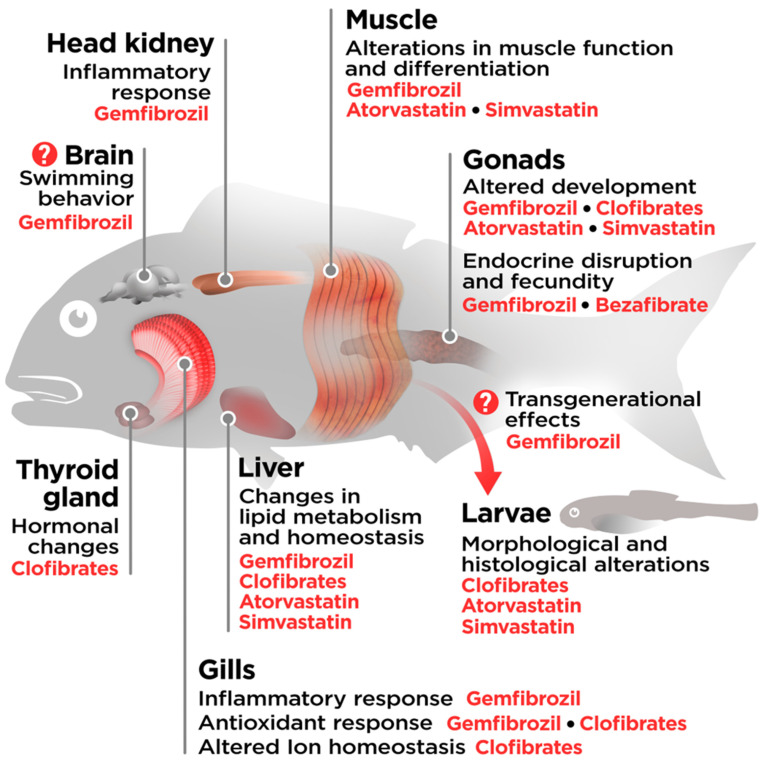
Putative effects of fibrates and statins in fish. Scheme of the effects of acute or chronic exposure to lipid regulators (see text for details) on major organs and tissues of fish.

**Table 1 animals-13-00792-t001:** The published literature on fibrates (gemfibrozil, clofibrate, clofibric acid and bezafibrate) and their main effects on fish.

Species	Drug	Exposure Period	Concentration	Exposure Route	Sample Type	Main Effects	Reference (2011–2022)
*Oncorhynchus mykiss*	GEM	15 d	100 mg/kg	Injection (IP)	Plasma, liver, adipose tissue, red muscle and white muscle	↓ lipoprotein concentration ↓ n-3 fatty acids	[45]
*Pimephales mykiss*	GEM	2, 8 and 21 d	1, 5, 15, 600, 1500 µg/L	Waterborne	Blood, liver, gonads, brain and pituitary	↓ plasma cholesterol at 600 µg/L in both sexes ↑ plasma Chol at 1500 µg/L in males ↑ *pparα* after 8 d at 600 µg/L in males ↑ *lpl* after 2 d at 600 µg/L in males ↓ *ldlr*, *fasn*, *apoeb, apoa1* in males ↓ *fasn*, *apoeb* in females ↓ *apoa1* at 2 d in females ↑ *apoa1* at 21 d in females	[48]
*Danio rerio*	GEM	6 w	0.5 and 10 μg/L	Waterborne	Plasma, gonads and whole body	↓ embryo viability ↓ survival in exposed embryos from non-exposed parents ↑ yolk-sac oedema ↑ skeletal deformities ↑ histopathological score ↑ regressive changes in kidney tubule	[67]
*D. rerio*	GEM	6 w	0.5 and 10 μg/L	Waterborne	Plasma, gonads and whole body	↓ embryo production ↑ embryo mortality after direct exposure ↑ histopathological score ↑ regressive changes in kidney tubule ↑ regressive changes in liver ↑ irregular oocytes in females	[68]
*Anguilla anguilla*	GEM	24 and 96 h	0.1, 1, 10, 2, 20, 200 μg/g BW	Injection (IP)	Blood and liver	↓ body weight ↓ EROD activity (CYP1A-like) ↓ BFCOD activity (CYP3A-like) ↓ CYP2K-like activity ↑ Hepatic AOX and CAT	[69]
*D. rerio*	GEM	6 w	0, 10 g/L	Waterborne	Embryos	↓ breeding success ↓ courtship time ↑ courtship frequency ↓ sperm size ↑ sperm speed	[70]
*Solea senegalensis*	GEM	2 d	1 mg/kg BW	Injection (IP)	Blood and liver	↑CYP450-related activities ↑ UDPGT ↓ hepatic CAT, GPx and GST.	[71]
*D. rerio*	GEM	30 d	16 µg/g BW	Diet	Liver and brain	↓ Chol ↓ TG, and E ↓ T in females ↑ PPARα in females ↑ PPARϒ ↑ liver SREBP-2, CYP3A65, atrogin1	[44]
*Sparus aurata*	GEM	96 h	0, 1.5, 15, 150, 1500 and 15,000 μg/L	Waterborne	Liver and blood	↑ APOA1, LPL ↑ IL-1β, TNF-α, and CASP3 genes at 15 µg/L. ↑ cortisol	[72]
*D. rerio*	GEM	120 and 144 h	0, 1.5, 3 and 6 mg	Waterborne	Embryos	↓ embryo and larvae survival ↑ yolk-sac deformities and oedema ↓ locomotion	[73]
*S. aurata*	GEM	96 h	1.5, 15, 150, 1500 and 15,000 μg/L	Waterborne	Blood	↑ DNA damage ↑cortisol at 1.5 μg L−1	[74]
*D. rerio*	GEM	42 d/67 d	0 and 10 μg/L	Waterborne	Blood, gonads, and whole bodies	↓ reproductive output ↓ whole body, plasma and testicular 11-KT	[75]
*D. rerio*	GEM	6 w	0 or 10 µg/L	Cell culture	Gonads	↓ sexual differentiation in F1 offspring depending on which parent was exposed	[76]
*S. aurata*	GEM	96 h	0, 1.5, 15, 150, 1500 and 15,000 μg/L	Waterborne	Gills and head kidney	↑ TOS, GPx1, and IL1β in gills and head kidney	[49]
*S. aurata*	GEM	96 h	1.5, 15, 150, 1500 and 15,000 μg/L	Waterborne	Liver, gills, brain and muscle	↓ locomotion capacity. ↑ CAT, GR, GPx ↑ TBARS ↓ LPO ↑ activity of enzymes of antioxidant defence activity (15–15,000 μg L−1).	[47]
*D. rerio*	GEM	6 w	10 μg/L	Water	Sperm, whole fish, blood and organs	↓ embryo production ↓ 11-KT ↓ courtship duration, ↓ sperm size ↑ sperm speed effects in unexposed F1, but not subsequent generations	[77]
*Oryzias latipes*	GEM	155 d (embryos) and 1 d (adults)	0, 0.04, 0.4, 3.7 and 40 mg/L	Waterborne	Blood, liver and gonads	↓ T at long-term exposure ↓ Hatching rate in F1 ↑ ER and VTG ↓ E2 and 11-KT at short-term exposure ↓ Chol in males	[78]
*Cyprinus carpio*	GEM, CLO and CA	-	1 mM	In vitro	Gonads	GEM and CLO: ↓ CYP11b CA: ↑ CYP11b CLO: ↓ CYP17	[79]
*D. rerio*	CLO	4 w	0, 0.25, 0.50, 0.75 and 1.00% of the diet	Diet	Liver, intestine, muscle, brain and heart	↑ FABP1, FABP7, FABP10, FABP11 ↑ acox1 ↑ peroxisomes (liver) and mitochondria (heart)	[80]
*Ctenopharyngodon idellal*	CLO	4 w	0, 1.25 g/kg	Diet	Liver, muscle and mesenteric fat	↓ TG, cholesterol and total lipid concentration ↑ hepatic ACO and LPL ↓ in FAS and ACC	[81]
*C. carpio*	CA	96 h	1, 10 and 100 μg/L	Waterborne	Blood	↓ RBC, plasma Na^+^, K^+^ and GOT ↑ WBC, glucose, protein, LDH enzyme and gill Na^+^/K^+^ ATPase	[82]
*Cirrhinus mrigala*	CA	96 h and 35 d	1, 10 and 100 μg/L	Waterborne	Blood	↓ plasma GOT, GPT and gill Na^+^/K^+^ ATPase at short-term exposure ↑ gill Na^+^/K^+^ ATPase at long-term exposure	[83]
*C. mrigala*	CA	96 h and 35 d	1, 10 and 100 μg/L	Waterborne	Blood	↑ plasmatic glucose, Na^+^ and K^+^ level at short-term exposure ↓ plasma protein and Cl levels ↑ plasma Na^+^ and Cl levels at long-term exposure	[84]
*C. mrigala*	CA	96 h and 35 d	1, 10 and 100 μg/L	Waterborne	Blood	↓ TSH ↓ T4 level at 1 and 100 g L−l at 96 h ↓ T4 at long-term exposure ↓ T3	[85]
*D. rerio*	CA	5 to 140 d	1 and 10 mg/g	Diet	Muscle, gonads and liver	↓, growth of F1 generation, TG in muscle and fecundity at high doses ↑ embryo’s abnormalities ↑ PPARα, PPARβ and ACO transcript of F1 ↑ weight of F2 generation (control diet) compared to F1 at low dose	[86]
*C. carpio*	CA	4 and 10 d	20 mg/L and 4 μg/L	Waterborne	Blood	↑ PPAR ↑Acox1 ↑ CYP4 and lpl, apoa1 ↓ CYP27A ↑ CYP2K, CYP3A, GSTP, MDR1, MRP2	[60]
*D. rerio*	CA	3, 6, 24, 48, 72 and 96 h	50 mg/L	Waterborne	Embryos	embryos present high metabolic potential to CA, with 18 transformation products detected	[87]
*D. rerio*	CA	120 h and 60 d	10.35, 20.7, 41.4, 82.8, and 165.6 μg/L	Waterborne	Eggs, embryos and adults	↑ SOD, GPx and LPO at high doses. ↑ CAT and GST at low dose ↓ CAT and GST at high dose ↓LPO at low dose ↓ total distance travelled and swimming time	[88]
*D. rerio*	BEZ	48 h, 7 and 21 d	1.7, 33 and 70 mg/g	Diet	Blood and gonads	↓ Chol ↓ 11-KT after 21 d ↓ in PPARβ and PPARϒ after 48 h ↑ PPARϒ PPARβ, StAR and CYP17A1 at 70 mg/g after 21 days. ↓ CYP19A1a ↑ cystic spermatocytes	[89]

Abbreviations: 11-ketotestosterone (11-KT); Acetyl coenzyme A carboxylase (ACC); Acyl-CoA oxidase activity (ACO, Acox); Apolipoprotein AI (APOA1); Bezafibrate (BEZ); Chloride (Cl); Clofibrate (CLO); Clofibric acid (CA); Days (d); Estrogen receptor (ER); Gemfibrozil (GEM); Glutamate oxaloacetate transaminase (GOT); Glutathione peroxidase 1 (GPx1); Glutathione S-transferase P (GSTP); Hours (h); Interleukin 1β (IL1β); Intraperitoneal Injection (IP); Lactate dehydrogenase (LDH); Lipoprotein lipase (LPL); Potassium (K^+^); Red blood cell (RBC); Sodium (Na^+^); Testosterone (T); Thyroid stimulating hormone (TSH); Thyroxine (T4); Triiodothyronine (T3); Vitellogenin (VTG); Weeks (w); White blood cell (WBC); Peroxisomal proliferation-associated receptor (PPAR); Peroxisomal proliferation-associated receptor α (PPARα); Peroxisomal proliferation-associated receptor ϒ (PPARϒ); Peroxisomal proliferation-associated receptor β (PPARβ) Glutamate oxaloacetate transaminase (GOT); Glutathione peroxidase (GPx); Farnesyl diphosphate synthase (FDPS); Superoxide dismutase (SOD); Sulphotransferase (sult2b); Glucose transporters (glut1b); Catalase (CAT); UDP-glucuronosyl transferase (UDPGT); Tumor necrosis factor α (TNFα); Steroidogenic Acute Regulatory Protein (StAR); Total oxidant status (TOS); Lactate dehydrogenase (LDH); Glutamate-pyruvate transaminase (GPT); Alternative oxidase (AOX); Cytochrome P450 (CYP450); Cytochrome P450, family 11, subfamily B (CYP11B); Cytochrome P450, family 17, subfamily A, polypeptide 1 (CYP17A1); Cytochrome P450, 1A (CYP1A); Cytochrome P450, 3A (CYP3A); Cytochrome P450, 2K (CYP2K); Cytochrome P450, family 3, subfamily A, polypeptide 65 (CYP3A65); Cytochrome P450, family 4 (CYP4); Cytochrome P450, family 27, subfamily A (CYP27A); Cytochrome P450, family 19, subfamily A, polypeptide 1a (CYP19A1a); 17β-estradiol (E2); Estradiol (E); Triglycerides (TG); sterol regulatory element-binding protein 2 (SREBP2); Caspase 3 (CASP3); Glutathione reductase (GR); Multidrug Resistance Mutation 1 (MDR1); Multidrug Resistance-Associated Protein 2 (MRP2); and Cholesterol (Chol).

**Table 2 animals-13-00792-t002:** The published literature on statins (atorvastatin and simvastatin) and their main effects on fish.

Species	Drug	Exposure Period	Concentration	Exposure Route	Samples Type	Main Effects	Reference (2012–2022)
*Oncorhynchus mykiss*	ATV	7 d	200 ng/L and 10 μg/L	Waterborne	Gills and liver	↑ SOD, MT, BAX, P-gp, MRP1 and SULT2b in gill at low concentration	[111]
*Danio rerio*	ATV	30 d	16 µg/g BW	Diet	Liver and brain	↓ cortisol, Chol, T, E, and atrogin1 ↑ TG and CYP3A65 in males ↓ TG in females ↑ PPARα in females ↑ PPARϒ, SREBP1, SREBP2, HMGCR1 and HMGCR2	[44]
*D. rerio*	ATV	120 h	0.045, 0.5, 1 mg/L	Waterborne	Larvae	↑ mortality ↓ spontaneous movement and response to physical stimuli ↓ COX, CS and LDH ↑ atrogen-1 at 0.045 mg/L and 1 mg/L ↓ atrogen-1 at 0.5 mg/L ↑ MURF ↑ *pgc-lα* at 0.045 mg/L ↓ *pgc-lα* at 0.5 mg/L and 1 mg/L	[112]
*O. mykiss*	ATV	3 or 6 h	45 ng mL−1 and 80 nM	In vitro	Hepatocytes	↑ HMGCR1, LDLR, PPARα, PPARϒ, and SREBP1 (lipid metabolism) ↑ CYP3A27 (xenobiotic metabolism) ↓ Chol	[113]
*D. rerio*	SIM	18 or 13 h	0.3, 3 and 6 nM, 0.375, 0.5 and 0.75 μM	Waterborne	Embryos	↑ mortality ↓ movement and heart rate uneven distribution of septa ↓ myofibril size ↓ somite size	[114]
*D. rerio*	SIM	18 or 13 h	0.3, 3 and 6 nM, 0.375, 0.5 and 0.75 μM	Waterborne	Embryos	↑ mortality ↓ chol ↓ yolk extension, tail length, somite length, septa angle ↓ swimming capacity ↑ pericardial oedema ↓ heartbeat ↓ total cell number	[115]
*D. rerio*	SIM	80 h	5 and 50 μg/L	Waterborne	Embryos	↑ mortality, tail abnormalities, developmental delays, pericardium oedema ↓ ABC proteins ↑ EROD and GST ↓ Cu/ZnSOD and CAT ↑ accumulation ↑ ABCB4, ABCC1, ABCC2, ABCG2A, CYP3A65, CYP1A1 and CAT at 50μg/l ↓ GST, SOD and CAT at 5 μg/l	[116]
*D. rerio*	SIM	2 and 80 h	5 or 50 μg/L	Waterborne	Embryos	↑ mortality ↑ PPARα, PPARβ, PPARϒ, PXR, AhR, RAR-α, RAR-β, RAG-α after 2 h exposure ↓ rxraa, rxrab, rxrbb, rxrga, rxrgb after 2 h exposure ↑ RAR-αb, and AhR. After 80 h exposure ↓ PXR and rxrgb after 80 h exposure	[117]
*D. rerio*	SIM	90 d	8, 40, 200 and 1000 ng/L	Waterborne	Liver	↓ body weight ↑ fertilization rate ↑ abnormalities (short tail, pericardium oedema, yolk-sac and eye deformities) ↓ heartbeat after parental exposure ↓ Chol and TG ↓ HMGCRA ↓ CYP51 in males	[118]
*Gambusia affinis*	SIM	24, 72 and 168 h	0.5, 5, 50 and 500 μg/L	Waterborne	Liver	↑ PXR, CYP3A, P-gp, MRP2, UGT ↓ PXR and MRP2 at high doses ↑ ERND (72 h) ↓ ERND (24) ↑ hepatocyte abnormalities at 5 μg/L1 (168 h)	[119]
*Mugilogobius abei*	SIM	72 h	0.5, 5, 50 and 500 μg/L	Waterborne	Liver	↑ P-gp, CYP1A, CYP3A, α-GST and PXR genes ↑ miR-148a at 50 μg/L ↑ miR-34c at 0.5μg/L (24 h) ↓ miR-34b and miR-34c at 0.5μg/L, 5μg/L, and 500μg/L (72 h) ↓ miR-34a at 500μg/L ↑ miR-27b at low doses ↓ miR-27b at high doses ↓ miR-27a ↑ ERND, T-SOD, CAT, GPX, MDA ↓ GSH ↓ hepatic vacuole diameter	[120]
*D. rerio*	SIM	90 d	8, 40, 200 and 1000 ng/L	Waterborne	Brain	↑ glut1b and COX4I1 in females ↓ glut1b and COX4I1 in males ↓ GAPDH ↓ ACADM and COX5aa in females ↑ ACADM in males	[121]
*G. affinis*	SIM	3 or 7 d	0.5, 5, 50 and 500 µg/L	Waterborne	Liver	↑ Nrf2 ↑ SOD2, CAT and GST ↑ GSTA (168 h) ↓ GCLC and GSTA ↓ NQO1 ↓ GSH ↑ GSH at 5 µg/L (24 h) ↓ MDA ↑ MDA (24 h) ↓ ERK ↓ JNK at low doses ↑ JNK at high doses ↑ histological changes in hepatic tissue (endoplasmic reticulum and mitochondria anomalies)	[122]
*D. rerio*	SIM	120 h and 60 d	92.45, 184.9, 369.8, 739.6 and 1479.2 ng/L	Waterborne	Embryos	↑ and ↓ locomotion (erratic vs. purposeful swimming, distance travelled and time) depending on dose and day/night phase ↑ Cu-ZnSOD ↓ GPx, GST, TBARS	[123]

Abbreviations: Atorvastatin (ATV); Simvastatin (SIM); Superoxide Dismutase (SOD); Metallothionein (MT), bcl-2 associated X protein (BAX), P-glycoprotein (P-gp), multidrug resistance protein 1 (MRP1), Sulfotransferase Family 2B (SULT2b); cholesterol (Chol), testosterone (T); estrogen (E); triglycerides (TG); cytochrome P450, family 3, subfamily A, polypeptide 65 (CYP3A65); cytochrome P450, family 3, subfamily A (CYP3A); cytochrome P450, family 3, subfamily A, polypeptide 27 (CYP3A27); cytochrome P450, family 1, subfamily A, polypeptide 1 (CYP1A); cytochrome P450, family 1, subfamily A, polypeptide 1 (CYP1A1); cytochrome P450, family 51 (CYP51); Peroxisome proliferator-activated receptor alpha (PPARα); Peroxisome proliferator-activated receptor Beta (PPARβ) peroxisome proliferator-activated receptor gamma (PPARϒ); sterol regulatory element-binding protein 1 (SREBP1); sterol regulatory element-binding protein 2 (SREBP2); 3-hydroxy-3-methyl-glutaryl-coenzyme A reductase 1 (HMGCR1); 3-hydroxy-3-methyl-glutaryl-coenzyme A (HMGCRA); 3-hydroxy-3-methyl-glutaryl-coenzyme A reductase 2 (HMGCR2); cytochrome oxidase (COX); citrate synthase (CS); lactate dehydrogenase (LDH); muscle RING-finer (MURF); peroxisome proliferator-activated receptor gamma coactivator 1α (pgc-1α); low-density lipoprotein receptor (LDLR); ATP-binding cassette transporter (ABC); ATP-binding cassette, subfamily B, member 4 (ABCB4); ATP-binding cassette, subfamily C, member 1 (ABCC1); ATP-binding cassette, subfamily C, member 2 (ABCC2); ATP-binding cassette, subfamily G, member 2A (ABCG2A); ethoxyresorufin-O-deethylase (EROD); glutathione S-transferases (GST); Copper (Cu); Zinc (Zn); catalase (CAT); pregnane X receptor (PXR); aryl hydrocarbon receptor (AhR); retinoic acid receptor alpha (RAR-α); retinoic acid receptor Beta (RAR-β); retinoic acid receptor alpha b (RAR-αb); recombination activating gene alpha (RAG-α); retinoid x receptor alpha a (rxrαa); retinoid x receptor alpha b (rxrαb); retinoid x receptor beta b (rxrβb); retinoid x receptor gamma a (rxrga); retinoid x receptor gamma b (rxrgb); multidrug resistance-associated protein 2 (MRP2); uridine 5′-diphospho-glucuronosyltransferase (UGT); Erythromycin-N-Demethylase (ERND); microRNA 148a (miR-148a); microRNA 148a (miR-148a); microRNA 34a (miR-34a); microRNA 34c (miR-34c); microRNA 34b (miR-34b); microRNA 27a (miR-27a); microRNA 27b (miR-27b); malondialdehyde (MDA); glutathione (GSH); Glutathione peroxidase (GPx); glucose transporter 1b (glut1b); cytochrome c oxidase subunits 4I1 (COX4I1); cytochrome c oxidase subunits 5aa (COX5aa); glyceraldehyde 3-phosphate dehydrogenase (GAPDH); medium-chain acyl-CoA dehydrogenase (ACADM); NF-E2-related factor 2 protein (Nrf2); superoxide dismutase 2 (SOD2); Alpha-glutathione S-transferase (GSTA); glutamate cysteine ligase catalytic subunit (GCLC); NAD(P)H quinone dehydrogenase 1 (NQO1); extracellular regulated protein kinase (ERK); c-Jun N-terminal kinase (JNK); and Thiobarbituric acid reactive substances (TBARS).

## Data Availability

Data sharing not applicable. No new data were created or analyzed in this study.

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
