# Peer review of "Elucidating the Effects of the Lipids Regulators Fibrates and Statins on the Health Status of Finfish Species: A Review"

_animals, 2023, doi:10.3390/ani13050792_

Round 1

Reviewer 1 Report

Manuscript Title:   Effects of fibrates and statins on wild and cultured fish.

Authors: Manuel Blonç, Jennifer Lima, Juan Carles Balasch, Lluis Tort, Carlos Gravato, Mariana Teles

The subject of this paper is a comprehensive review on the effects of fibrates and statins on fish.

This work helps to understand an environmental problem which is less studied and where few data are available. It has an acceptable originality and correctness and the importance to the field is adequate and of broad international interest. The paper is well written. The organization and clarity of this work is very good and is described comprehensively. In general, the review is well supported by data presented. This study is within the scope of the journal.

Minor Revisions:

L140: correct ng/L-1 to ng/L

L159: correct to ng/L

Figure 1: I suggest changing to “Figure 1. Putative effects of fibrates and statins in fish. Scheme of the effects of acute or chronic exposure to lipid regulators (see text for details) on major organs and tissues of fish”.

Table 1 and Table 2: some corrections are required: uniformity of the units – by using throughout the manuscript …/L instead of -1 the table must be accordingly.

For Pimephales mykiss where is ug/L it must be µg/L

Author Response

Answer to reviewer #1’s comments

Animals, Manuscript ID: animals-2184036

Title: Effects of fibrates and statins on wild and cultured fish

  • New title (As suggested by reviewer #2): Elucidating the effects of the Lipids Regulators Fibrates and Statins on the Health Status of Finfish Species: A review.

First of all, thank you for considering our manuscript for publication in this journal. We have

addressed all the reviewer’s suggestions, hoping that our changes can provide adequate

answers to them all. We sincerely thank the reviewer for their suggestions, as they contributed to

improving the quality of the manuscript. Following are the comments from reviewer #1addressed one by one.

REVIEWER #1:

The subject of this paper is a comprehensive review on the effects of fibrates and statins on fish.

This work helps to understand an environmental problem which is less studied and where few data are available. It has an acceptable originality and correctness and the importance to the field is adequate and of broad international interest. The paper is well written. The organization and clarity of this work is very good and is described comprehensively. In general, the review is well supported by data presented. This study is within the scope of the journal.

Answer: Thank you for your comment.

Minor Revisions:

Comment 1: L140: correct ng/L-1 to ng/L

Answer: The change has been made accordingly

Comment 2: L159: correct to ng/L

Answer: The change has been made accordingly. All errors of this type have been corrected in the text.

Comment 3: Figure 1: I suggest changing to “Figure 1. Putative effects of fibrates and statins in fish. Scheme of the effects of acute or chronic exposure to lipid regulators (see text for details) on major organs and tissues of fish”.

Answer: The Figure legend has been changed as suggested by Reviewer #1.

Comment 4: Table 1 and Table 2: some corrections are required: uniformity of the units – by using throughout the manuscript …/L instead of -1 the table must be accordingly.

Answer: The units have been uniformed throughout the tables as recommended by the reviewer.

Comment 5: For Pimephales mykiss where is ug/L it must be µg/L

Answer: The units have been corrected and standardised in all similar cases.

Reviewer 2 Report

The review article with ID (animals-2184036) by Blonç and coauthors is fascinating; however, it needs extensive revisions in order to be accepted for publishing in Animals. Authors have to pay more time and work to improve it as they can. Also, authors should prepare a point-by-point response to the comments and criticisms raised by the anonymous reviewer.

Title:

I suggest removing “on wild and cultured fish” from the title. My suggestion is to change the title to be

“Elucidating the effects of the Lipids Regulators Fibrates and Statins on the Health Status of Finfish Species: A review”

The authors are free to use or not. I suggest only using attractive and subject-related title to attract readers.

Simple Summary:

Line 11: The word “Medicine” is incorrect here. You can use “Medicinal pharmaceuticals” or “Pharmaceuticals”

Lines 16-17: “species, commonly produced by the European aquaculture industry”. This information is also incorrect. This is because the finfish species included in this review article are not only related to the “European aquaculture industry”. Thus, you should remove this sentence and replace it with “several finfish species worldwide”

Lines 18-19: you may use “causing reproductive and developmental disorders” which will be more appropriate here.

Line 20: The word “extremely” should be removed from this site, in the “abstract” and also from the conclusion also. This is because you already presented several references on the subject of your review article.

Abstract:

Lines 23-24: The first line is already present in the “Simple Summary”. Thus, there is no need to repeat it.

Lines 26-27: These two lines are also found in the “Simple Summary”. Thus, there is no need to repeat them.

Line 37: Remove “is still extremely limited”

Lines 38-39: “aquaculture production, global food security and, ultimately, human health”. This sentence is also present in “Simple summary”. Avoid repetitions not to let the readers get bored. Short information and directions to the goals are more interesting than long and repeated ones.

1. Introduction

Line 43: Remove “lipid regulators in aquatic and cultured systems” from the title. This is because most of the information present underneath not only focused on lipid regulators but also reviewed several other issues.

Line 48: “surface waters, and within organisms” – also add “surface, water column, ground waters, and within organisms”

Line 53: are not monitored

Line 59: “the over and misuse of pharmaceuticals,

Lines 66-99: These 33 lines focused mainly on the RAS system. I personally do not know why the authors included these lines in the text. I found that you should concentrate your lines and knowledge on the subject of your review article. I found that focusing on “lipid regulators” is very important and avoids distracting readers' thoughts.

Line 114: Authors should write on the main objectives of this review article.

A general notice in the review text: -

I found that authors only review the literature cited according to the previously published papers. Two points are missed here. The authors have to add their input and also have to write their opinion on the published papers. The authors should write some conclusions and perspectives. It is not logical that authors review the published papers only. Write in your own words and classify the effect of each one of the lipid regulators on fish. Please look at the presented figure 1. It would be best if you wrote the main effects produced by each one of the studied pharmaceuticals. Have they affected immunity, growth performance, antioxidant condition, serum biochemistry, fish behavior (swimming), and reproductive performances? Moreover, how these pharmaceuticals exert their modes of action to make these effects. Appropriate discussion MUST be included after each point. Do not write the findings of others ONLY. It would be best if you went in depth in each one.

Tables (Table 1 and Table 2)

General comments

§  Authors must use the MDPI reference style to write the references inside the tables. I suggest using EndNote or any other reference management program.

§  Abbreviate the Latin name of the fish species after its first appearance in the table.

§  Change “Exposure Pathway” to “Exposure route”

§  Authors must add the points in the “Main effects” column as bullet points. This is important to decrease and also concentrate the information toward the main results according to the published literature

§  My advice to the authors is to avoid using the same text used by the authors of the published papers, particularly in the column “Main effects”

§  I found that several other indices present in “Main effects” could be abbreviated. This notice could decrease the content in this column and avoid repetitions.

§  Authors also can use upward (↑) and downward (↓) arrows to replace the “increase” or “decrease” that is present in the main effects column.

Table 1

ü  Skolness et al., 2012 - Galus et al., 2013a - Galus et al., 2013b: The information present in the column “Main effects” in these references is too general. Authors should add specific information according to the published paper.

ü  Barreto et al., 2018: Remove “Different responses on gills and liver”

ü  Coimbra et al., 2015: At low dose, was found an in-crease of embryo's abnormalities in the offspring. This is too general information. Please revise all references in this manner to avoid the repetition of all my comments.

ü  Corcoran et al., 2015: “several biotransformation genes” – The same comment as above

ü  Brox et al., 2016: same comment

ü  Rebelo et al., 2020: “Behavior alterations” – same comment

ü  Velasco-Santamaría et al, 2011: “Alterations in gonadal histology” – same comment

Table 2

Please follow the comments raised by the reviewer as same as in general comments on tables and specific comments on Table 1.

4. Concluding remarks

Line 447: Remove “recognised as” and “extremely”

Line 461: Replace “are at risk” by “will be at risk”

Line 463: “additional research” – I found that authors should add their input to this point. Authors should specify which issues will be needed to complete the knowledge about the effects of statins and fibrates on fish health and welfare. Authors should illustrate concise points that will be required for or warrant further additional investigations. These points may shed light for future researchers to complete.

References

This section contains minor revisions regarding the writing of the journal names.

Line 600: Integr. Pharm. Res. Pract.

Line 608: Nat. Clin. Pract. Cardiovasc. Med.

Author Response

ANSWER TO REVIEWER #2’S COMMENTS:

Animals, Manuscript ID: animals-2184036

Title: Effects of fibrates and statins on wild and cultured fish

  • New title (as suggested by reviewer #2): Elucidating the effects ofthe Lipids Regulators Fibrates and Statins on the Health Status of Finfish Species: A review.

First of all, thank you for considering our manuscript for publication in this journal. We have addressed all the reviewer’s suggestions, hoping that our changes can provide adequate answers to them all. We sincerely thank the reviewer for their comments, as they contributed to improving the quality of the manuscript. Following are the comments from reviewer #2, addressed one by one.

REVIEWER #2:

The review article with ID (animals-2184036) by Blonç and coauthors is fascinating; however, it needs extensive revisions in order to be accepted for publishing in Animals. Authors have to pay more time and work to improve it as they can. Also, authors should prepare a point-by-point response to the comments and criticisms raised by the anonymous reviewer.

We want to thank Reviewer #2 for their comments. They significantly increase the quality of the present review. A point-by-point response to the comments follows below.

Title:

Comment 1: I suggest removing “on wild and cultured fish” from the title. My suggestion is to change the title to be “Elucidating the effects of the Lipids Regulators Fibrates and Statins on the Health Status of Finfish Species: A review”

The authors are free to use or not. I suggest only using attractive and subject-related title to attract readers.

Answer: The title has been modified as suggested by the reviewer.

Simple Summary:

Comment 2: Line 11: The word “Medicine” is incorrect here. You can use “Medicinal pharmaceuticals” or “Pharmaceuticals”

Answer: The word “Medicine” has been changed by “Pharmaceuticals” as suggested. (Line 14).

Comment 3: Lines 16-17: “species, commonly produced by the European aquaculture industry”. This information is also incorrect. This is because the finfish species included in this review article are not only related to the “European aquaculture industry”. Thus, you should remove this sentence and replace it with “several finfish species worldwide”

Answer: The suggested changes have been made (Line 18-19).

Comment 4: Lines 18-19: you may use “causing reproductive and developmental disorders” which will be more appropriate here.

Answer: The sentence has been changed as recommended. (Line 22).

Comment 5: Line 20: The word “extremely” should be removed from this site, in the “abstract” and also from the conclusion also. This is because you already presented several references on the subject of your review article.

Answer: The word “Extremely” has been removed from the simple summary (line 24), the abstract (line 41), and the concluding remarks (line 511).

Abstract:

Comment 6: Lines 23-24: The first line is already present in the “Simple Summary”. Thus, there is no need to repeat it.

Answer: The first sentence of the abstract has been deleted as suggested. (Line 27-28)

Comment 7: Lines 26-27: These two lines are also found in the “Simple Summary”. Thus, there is no need to repeat them.

Answer: The sentence has been removed. (Line 30-32).

Comment 8: Line 37: Remove “is still extremely limited”

Answer: The word “extremely” has been removed (Line 41).

Comment 9: Lines 38-39: “aquaculture production, global food security and, ultimately, human health”. This sentence is also present in “Simple summary”. Avoid repetitions not to let the readers get bored. Short information and directions to the goals are more interesting than long and repeated ones.

Answer: The sentence has been simplified in the “Simple summary” (Line 26) in order for it to remain as it currently is in the abstract (Line 42-43).

Introduction

Comment 10: Line 43: Remove “lipid regulators in aquatic and cultured systems” from the title. This is because most of the information present underneath not only focused on lipid regulators but also reviewed several other issues.

Answer: The Heading has been changed to “Introduction” (Line 48).

Comment 11: Line 48: “surface waters, and within organisms” – also add “surface, water column, ground waters, and within organisms”

Answer: “Ground water” has been added to the sentence (line 53). However, in this case, surface water refers to water above ground (c.f. ground water), and not the surface of a body of water. For this reason, “water column” has not been specified.

Comment 12: Line 53: are not monitored

Answer: Change made. (Line 58).

Comment 13: Line 59: “the over and misuse of pharmaceuticals,

Answer: The wording has been modified as suggested. (Line 63-64).

Comment 14: Lines 66-99: These 33 lines focused mainly on the RAS system. I personally do not know why the authors included these lines in the text. I found that you should concentrate your lines and knowledge on the subject of your review article. I found that focusing on “lipid regulators” is very important and avoids distracting readers' thoughts.

Answer: One of the objectives of this work was to investigate the potential effect of lipid regulating agents in European RAS. For this reason, literature focusing on the effects of statins and fibrates on the most commonly farmed fish in RAS (e.g. Atlantic salmon, Rainbow trout, European eel) was given special consideration. Therefore, these lines were included to introduce the readers to the concept or RAS and its current status in the European Union.

Comment 15: Line 114: Authors should write on the main objectives of this review article.

Answer: The main objectives have been added at the end of section 1 “Introduction” (Page 3, lines 120-122)

A general notice in the review text: -

Comment 16: I found that authors only review the literature cited according to the previously published papers. Two points are missed here. The authors have to add their input and also have to write their opinion on the published papers. The authors should write some conclusions and perspectives. It is not logical that authors review the published papers only. Write in your own words and classify the effect of each one of the lipid regulators on fish. Please look at the presented figure 1. It would be best if you wrote the main effects produced by each one of the studied pharmaceuticals. Have they affected immunity, growth performance, antioxidant condition, serum biochemistry, fish behavior (swimming), and reproductive performances? Moreover, how these pharmaceuticals exert their modes of action to make these effects. Appropriate discussion MUST be included after each point. Do not write the findings of others ONLY. It would be best if you went in depth in each one.

Answer: General comments on the reviewed findings, as well as a summary on the general effects these compounds have on fish have been added (Lines 355-394; 493-504). Throughout the text, comments on specific publications have been given. In our opinion, going into depth on the modes of actions of each compound, and how these relate to the observed effects would significantly increase the length of the review and become very speculative. In addition, the present work focuses on the main effects of lipid regulators in fish, rather than providing a complete biochemical review.

Tables (Table 1 and Table 2)

General comments

Comment 17: §  Authors must use the MDPI reference style to write the references inside the tables. I suggest using EndNote or any other reference management program.

Answer: Done. All references in the tables now follow the MDPI reference style.

Comment 18: §  Abbreviate the Latin name of the fish species after its first appearance in the table.

Answer: Done. The full scientific name has been displayed the first time each species appears in each table, after which only the abbreviated latin name has been employed.

Comment 19: §  Change “Exposure Pathway” to “Exposure route”

Answer: Done.

Comment 20: §  Authors must add the points in the “Main effects” column as bullet points. This is important to decrease and also concentrate the information toward the main results according to the published literature

Answer: Done. Information displayed in the “main effects” column has been changed to bullet points in both tables.

Comment 21: §  My advice to the authors is to avoid using the same text used by the authors of the published papers, particularly in the column “Main effects”

Answer: Done. The information displayed in the column “Main effects” has been reworded as to avoid plagiarism.

Comment 22: §  I found that several other indices present in “Main effects” could be abbreviated. This notice could decrease the content in this column and avoid repetitions.

Answer: The recommended change has been done, and abbreviations have been described below the table.

Comment 23: §  Authors also can use upward (↑) and downward (↓) arrows to replace the “increase” or “decrease” that is present in the main effects column.

Answer: increase/decrease and synonyms have been replaced by arrows as suggested.

Table 1

Comment 24: ü  Skolness et al., 2012 - Galus et al., 2013a - Galus et al., 2013b: The information present in the column “Main effects” in these references is too general. Authors should add specific information according to the published paper.

Answer: The information obtained from these articles has been detailed.

Comment 25: ü  Barreto et al., 2018: Remove “Different responses on gills and liver”

Answer: This sentence has been removed.

Comment 26: ü  Coimbra et al., 2015: At low dose, was found an in-crease of embryo's abnormalities in the offspring. This is too general information. Please revise all references in this manner to avoid the repetition of all my comments.

Answer: The information has been presented in a more detailed manner, taking into account previous comments from Reviewer #2 (e.g. abbreviations and bullet points).

Comment 27: ü  Corcoran et al., 2015: “several biotransformation genes” – The same comment as above

Answer: The information has been detailed.

Comment 28: ü  Brox et al., 2016: same comment

Answer: Further detail has been provided in the table.

Comment 29: ü  Rebelo et al., 2020: “Behavior alterations” – same comment

Answer: Details on the results from this article have been included in the form of bullet points.

Comment 30: ü  Velasco-Santamaría et al, 2011: “Alterations in gonadal histology” – same comment

Answer: Details on the results from this article have been included in the form of bullet points.

Table 2

Comment 31: Please follow the comments raised by the reviewer as same as in general comments on tables and specific comments on Table 1.

Answer: Table 2 has been corrected in the same manner as table 1, as suggested by the reviewer. Details have been provided on the results of each article as bullet points, with all possible abbreviations used, and avoiding plagiarism.

Concluding remarks

Comment 32: Line 447: Remove “recognised as” and “extremely”

Answer: Done. (Line 511).

Comment 33: Line 461: Replace “are at risk” by “will be at risk”

Answer: Done. (Line 553).

Comment 34: Line 463: “additional research” – I found that authors should add their input to this point. Authors should specify which issues will be needed to complete the knowledge about the effects of statins and fibrates on fish health and welfare. Authors should illustrate concise points that will be required for or warrant further additional investigations. These points may shed light for future researchers to complete.

Answer: The section “Concluding remarks” has been extended in order to match the reviewer’s suggestions (Lines 530-557).

References

This section contains minor revisions regarding the writing of the journal names.

Comment 35: Line 600: Integr. Pharm. Res. Pract.

Answer: Modification made. (Line 693).

Comment 36: Line 608: Nat. Clin. Pract. Cardiovasc. Med.

Answer: Modification made. (Line 701).

Round 2

Reviewer 2 Report

The authors properly addressed the comments raised by the anonymous reviewer. The manuscript merits acceptance in its current form